# Agricultural Machinery Adequacy for Handling the Mombaça Grass Biomass in Agroforestry Systems

**Gelton Fernando de Morais** [1,*] **, Jenyffer da Silva Gomes Santos** [1]**, Daniela Han** [2]**, Luiz Octávio Ramos Filho** [3]**, Marcelo Gomes Barroca Xavier** [4]**, Leonardo Schimidt** [5]**, Hugo Thiago de Souza** [2]**, Fernanda Ticianelli de Castro** [4]**, Vanilde Ferreira de Souza-Esquerdo** [1] **and Daniel Albiero** [1]

[1] Faculdade de Engenharia Agrícola, Universidade Estadual de Campinas, Campinas 13083-875, Brazil; j215986@dac.unicamp.br (J.d.S.G.S.); vanilde.esquerdo@feagri.unicamp.br (V.F.d.S.-E.); daniel.albiero@feagri.unicamp.br (D.A.)

[2] Faculdade de Ciências Agronômicas, Universidade Estadual Paulista, Botucatu 18610-034, Brazil; daniela.han@unesp.br (D.H.); hugo.souza@unesp.br (H.T.d.S.)

[3] Empresa Brasileira de Pesquisa Agropecuária, Jaguariúna 13918-110, Brazil; luiz.ramos@embrapa.br

[4] Centro de Ciências Agrárias, Universidade Federal de São Carlos, Araras 13600-970, Brazil; marceloxavier@estudante.ufscar.br (M.G.B.X.); fernandaticianelli@estudante.ufscar.br (F.T.d.C.)

[5] Instituto de Biologia, Universidade Estadual de Campinas, Campinas 13083-862, Brazil; l220179@dac.unicamp.br

\* Correspondence: gelton_morais@hotmail.com

**Abstract:** The current scenario of Agroforestry Systems (AFS) worldwide lacks specific machinery, resulting in practically all operations being carried out manually. This leads to a significant physical effort for small-scale farmers and limits the implementation of AFS to small areas. The objective of the study was to evaluate the suitability of existing machines for performing agroforestry tasks. This research utilizes Descriptive Statistics and Exponentially Weighted Moving Average methods to evaluate the data and compare the treatments, where different machines are used to cut Mombaça grass (*Megathyrsus maximus* Jacq): (i) costal brushcutter (CBC); (ii) tractor-mounted rotary brushcutter (RBC); and (iii) mini grain reaper machine (GRM). The experiments were conducted in Jaguariúna, São Paulo, Brazil. GRM is recommended for achieving greater biomass production, reducing raking time, and minimizing operational costs. CBC is suitable for smaller areas due to its affordability and slow operation, which requires significant physical effort. RBC is recommended for reducing working time, physical effort, and personnel costs, making it suitable for larger-scale contexts.

**Keywords:** agroforestry mechanization; agroforestry mechanization; *Megathyrsus maximus* Jacq; agri-machines; forest farming; interrow production; machine suitability



## 1. Introduction

The emergence of agriculture occurred approximately ten thousand years ago [1]. Agriculture allowed for population growth and social and cultural changes, making it essential for the development of the human species [2]. Modern agriculture developed during the 20th century, especially in the 1960s, also known as the Green Revolution. However, this agricultural model came with various negative impacts, such as increasing pesticide use, land expropriations, land concentration, and biodiversity reduction [3]. Furthermore, the employment of automated machinery in industrial farming on a vast scale creates unjust competition for market share with small-scale farmers. This is because the mechanization of the production process is directly linked to agricultural productivity and the provision of products to consumers [4].

In Brazil, the discussion surrounding agroecology has emerged as a challenge the traditional production model by advocating for changes in production techniques [5]. A prominent example is the Landless Workers' Movement (MST), which has actively

promoted and spread agroecology throughout its settlements as a strategic guideline since the mid-1990s [6]. The adoption of agroecological practices has resulted in reduced dependency on synthetic inputs in cropping systems [7]. Agroforestry Systems (AFSs) are based on principles similar to those of agroecological systems [8]: they are farming practices that promote biodiversity and enhance natural processes by mimicking the principles of natural forest ecosystems.

Introduced in 1977, the term "agroforestry" began to be used by researchers on integrated production systems combining crops and trees [9]. These systems involve the cultivation of perennial woody plants, as well as herbaceous, shrub, tree, agricultural, and forage crops in the same land area [10]. When compared to croplands, the effects on fauna abundance and diversity are positive in AFSs [11]. AFSs help in the restoration of degraded lands and also provide food, where trees composing these systems provide various ecosystem services directly related to environmental quality and the well-being of human populations [12]. AFSs play a fundamental role in diversifying farm production, sustaining crop yields, and ensuring environmental integrity in land use [9].

AFSs have immense potential to reduce emissions from agriculture by being able to sequester carbon from the atmosphere [13] in a range between 1.1 and 34.2 Pg C globally [14] as well as increasing soil carbon [15]. AFSs integrate systems to address both environmental and socio-economic objectives, preventing environmental degradation and soil erosion, combating climate change and biodiversity loss, reducing poverty, improving agricultural productivity, and promoting the well-being of soil and ecosystems by ensuring their health, while also simultaneously offering stable incomes, sustainable production, food security, diversified human diets, resilience, economic opportunities, and other advantages to farmers [13,15,16].

AFSs exhibit a remarkable level of complexity and biodiversity, which predominantly necessitates manual labor. Regrettably, manual work proves to be less efficient when compared to mechanized labor, thereby posing challenges in employing AFSs for large-scale agriculture. Moreover, the currently available mechanization options in the market are ill-suited for the intricate demands of AFSs, as they have been predominantly designed and optimized for monoculture operations [17,18]. Monoculture enables the more efficient utilization of farm machinery for cultivation, sowing, weed control, and harvesting [19].

In addition to the lack of dedicated mechanization, farmers encounter a range of additional challenges. These include inadequate income [20], high expenses [21], limited government incentives [14], insufficient availability of large land areas that warrant machinery investment [22], an aging population [20], adherence to traditional practices [23], and escalating population density and labor availability in urban areas [24]. Agricultural mechanization makes work easier and significantly reduces production costs [25] at the same time, and it is likely to have broader impacts on agronomy, the environment, and socioeconomic factors than is commonly recognized [26].

This project aims to tackle the issue of insufficient mechanization for AFSs by collecting essential data to facilitate the use of existing machinery for AFS operations. The information and recommendations on mechanization gathered through this study can be disseminated to farmers, extension technicians, and policymakers. This dissemination would contribute to streamlining the substantial workload associated with family farming, where labor is scarce, and where AFSs serve as a source of income. Specifically, the focus is on optimizing the management of grass biomass produced in the interrows of the AFS model examined in this study.

The AFS model discussed in this context was inspired by the models employed by the Programa Microbacias II—Subprojetos ambientais—PDRS, which was led by the Secretaria Estadual de Meio Ambiente (SMA-SP) in São Paulo in 2017. The interrows are planted with grass, while a variety of other species make up the rows. Managing the grass biomass produced in situ has the potential to enhance soil physical, chemical, and biological properties [27]. Utilizing conventional machines for interrow handling can significantly reduce the manual labor required, making the work less strenuous and

resulting in favorable outcomes, such as increased productivity and competitiveness in the market [25].

The premise of this study is that the quantity of grass biomass produced during each cutting differs depending on the machine used for the task. This study aims to identify the strengths and weaknesses of each tested machine, compare their performances, and determine the situations in which they are best suited in order to provide guidance in selecting the optimal machine for managing interrow biomass in AFSs. This has been achieved through the characterization of their work and comparative analysis of data using EWMA. This statistical control method was preferred because the data distribution was not normal, according to other authors who also used EWMA for statistical control of non-normal data about agricultural mechanization, such as [28–32].

## 2. Materials and Methods

In this research, three machines were assessed: two brushcutters that use blade impact for grass cutting one with a front cutting mechanism, named the costal brushcutter (CBC), and the other with a rear cutting mechanism, named the tractor-mounted rotary brushcutter (RBC), and, thirdly, a mini grain reaper machine (GRM) that is traditionally employed for harvesting rice and wheat. The inclusion of GRM was based on its superior front cutting system, which allows for the comparison of grass response to different cutting methods. Since each machine employs a unique cutting system, the regrowth of plants and subsequent biomass productivity may differ across cutting cycles.

To examine this hypothesis, the comparative approach of Exponentially Weighted Moving Averages (EWMA) was employed, as recommended by [33], as it is a statistical control method suitable for non-normal distributed data, as is the case in this investigation. The randomized block experimental design [34] was implemented to gather information on the analyzed variables. The experiment consisted of six blocks, each containing three plots representing a specific treatment. The grass was subjected to three cutting events over the course of the year-long study. The same grass species, Mombaça grass (*Megathyrsus maximus* Jacq), was cultivated throughout the experiment.

### 2.1. Experiment General Characterization, Implementation and Conduction

The experimental field of the Empresa Brasileira de Pesquisa e Agropecuária (Embrapa) was utilized in the experiments aimed at replicating interrow conditions of the AFS model investigated in this study. The AFS model consists of tree species of economic value planted in rows and interrows, which produce Mombaça grass biomass. The interrow areas play a crucial role in generating a maximum amount of biomass that, after harvest, serves as soil cover for the economically valuable tree species planted in rows. To simulate interrow areas of AFSs, the experiment involved cultivating grass plots with the same width as the interrows of AFSs, typically 6 m. Since the rows of economically valuable tree species were not planted, the experiment was conducted as a monoculture.

To carry out the experiment, the soil was prepared by using plow harrow, leveling harrow, and subsoiler. The application of natural phosphate fertilizer was undertaken prior to the sowing of grass, which was achieved manually at a density of 3 kg of pure viable seeds per hectare, following the sowing rate method (Equation (1)) described by [35].

$$SR = (PVS \times 100)/CV \qquad (1)$$

where SR = sowing rate (kg/ha); PVS = pure viable seeds recommended in the literature (kg/ha); and CV = cultural value, which expresses the PVS of a given seed lot (%).

The area for sowing was previously demarcated by plotting blocks, and the seeds were incorporated into the soil using a leveling harrow. After the grass had emerged and grown, a standardization cut was performed using a tractor mounted rotary brush cutter. All plots and blocks were cut using the same machine, and subsequent cuts were made based on the treatments.

### 2.2. Study Area

This research was conducted in the southeastern region of Brazil, specifically at Embrapa, located in Jaguariúna town in the state of São Paulo at coordinates 22°43′28.41″ S and 47°0′56.08″ W (Figure 1).

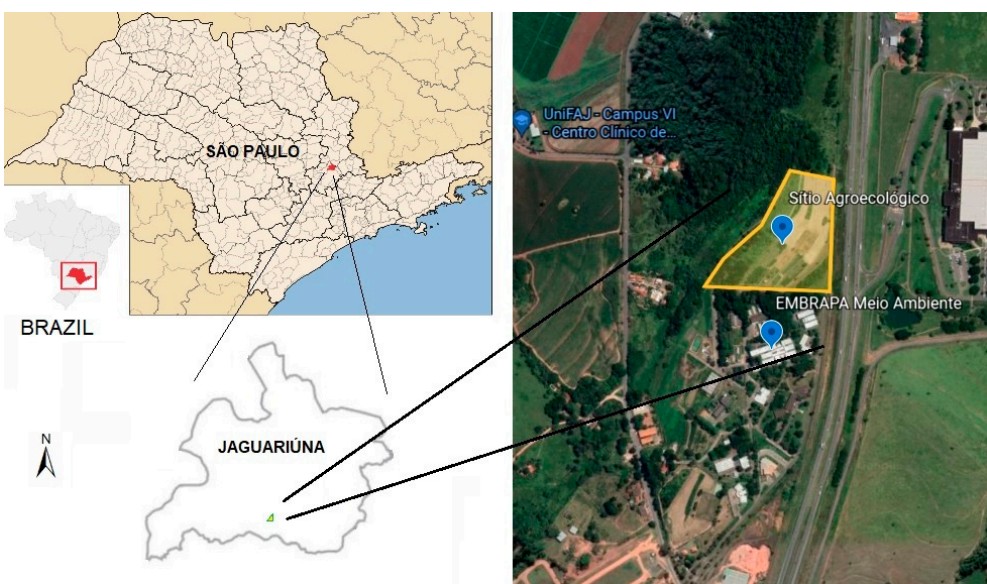

**Figure 1.** Embrapa location.

The experimental area (Figure 2) covered 5 hectares, as reported by [36]. The predominant soil in the area is classified as Dystrophic Red-Yellow Latosol, with a sandy-clay-loam texture and a moderate A-horizon. The region is also characterized by a subdeciduous tropical forest phase, according to the classification by [37].

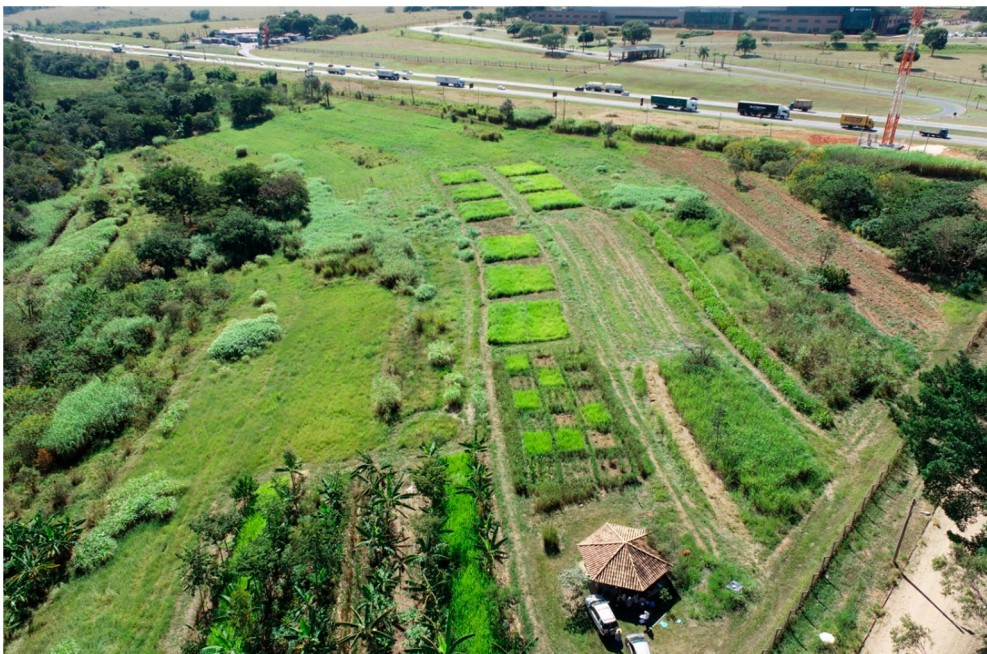

**Figure 2.** Experimental area.

### 2.3. Agroforestry System Studied

The focus of this study was on the mechanized handling of Mombaça grass cultivated in a specific type of Agroforestry System (AFS). In this AFS model (Figure 3), rows of tree species with economic value are planted, while the interrow space is populated with grass. This approach is commonly employed in degraded areas with exposed soil. The grass biomass is periodically harvested and utilized as a soil cover on the rows of tree species. This practice offers numerous advantages, including weed control without the use of herbicides, decreased temperature fluctuations, maintenance of soil moisture, and the provision of organic matter for soil organisms [38]. The presence of abundant soil biota leads to the continuous decomposition of organic matter, which, in turn, enriches the soil and supports plant growth.

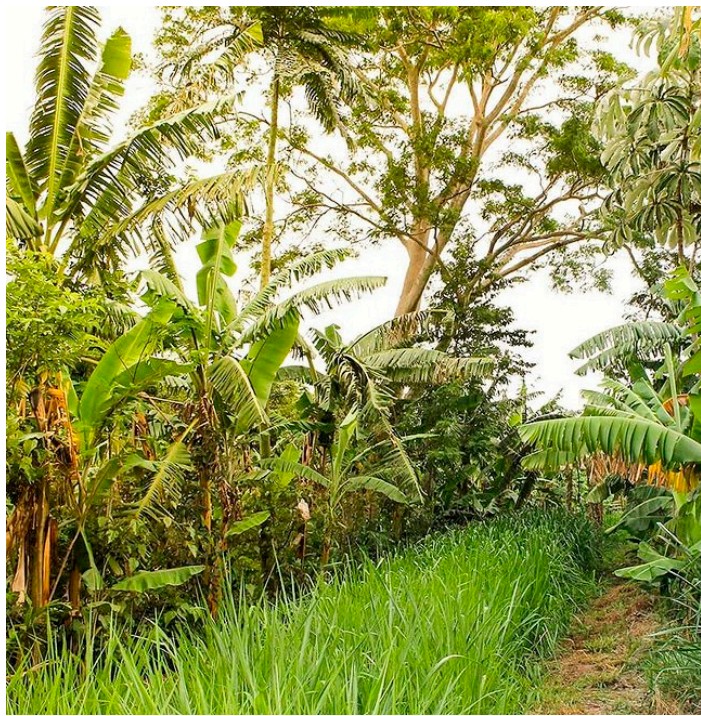

**Figure 3.** AFS model studied.

### 2.4. Material

#### 2.4.1. Cultivated Species

The grass variety chosen for the experiments in this study was *Megathyrsus maximus* Jacq, commonly known as Mombaça grass. This cultivar is frequently employed in Agroforestry Systems owing to its rapid and abundant biomass generation in comparison to other grass varieties [39]. Moreover, its biomass is abundant in carbon, which results in a gradual decomposition process, providing longer-lasting soil protection [40].

#### 2.4.2. Tested Machines

To compare the different handling techniques, three machinery options were chosen for testing and data generation: (i) costal brushcutter (CBC); (ii) tractor mounted rotary brushcutter (RBC); and (iii) mini grain reaper machine (GRM). CBC and RBC are commonly used in Brazil, and are readily available to small-scale farmers. GRM is imported and less accessible, but it offers the advantage of a gentler grass-cutting mechanism that minimizes damage to the grass tussocks. The selection of these machines was based on the hypothesis that optimizing agroforestry management with accessible machinery would be advantageous and that the GRM would lead to greater production of grass biomass due to its cutting system over successive cuts.

- Costal brushcutter. A portable machine (Figure 4a) with rotating blades (Figure 4b) driven by power that is generated by a power take-off (PTO). It incorporates a compact engine connected to a cutting disc, and it is secured to the operator's torso through a belt. The machine has a weight of approximately 14 kg, and it features a three-pointed cutting blade. The specific model tested in this study was the Husqvarna 143 R-II, which runs on gasoline, has a cylinder volume of 41.5 $cm^3$, a power output of 1.5 kW (2.01 hp), a maximum speed of 7500 rpm, and a fuel tank capacity of 0.95 L.

- Tractor mounted rotary brushcutter. The specific model utilized in the experiment was the Jumil JM-RUTD-A 1.4, manufactured in 2010 with a cutting width of 1.3 m and a cutting height that can be adjusted between 2 and 10 cm. It is equipped with two rotating blades (Figure 5a) that operate at a speed between 800 and 1.100 rpm, and it weighs 342 kg, with dimensions of 1.5 m in width, 1.95 m in length, and 1.13 m in height. An attachment can be connected to the three-point system of a tractor using a drawbar and hitch (Figure 5b). The machine is driven by the tractor's PTO, and it is specifically designed for cutting forage and controlling weeds and unwanted plants. The tractor employed to operate the brushcutter was a Tramontini T5045-4, which is a 4 × 4 model from the Brasil Cafeeiro series (Figure 5c). It has a width of 1.170 m, hydrostatic steering, hydraulic auxiliary control, a dual-stage clutch, PTO that can be set to 540 to 1000 rpm, 50 CV of power, and a 4-cylinder engine.

- Mini grain reaper machine. The machine was chosen for its superior cutting system, which minimizes damage to the grass tussocks and ensures their health for subsequent growth cycles. This is expected to result in enhanced regrowth vigor and increased biomass production during successive cuttings. The machine, used as a prototype in this study, is small in size and designed as a reaper-type harvester originally intended for grains like rice and wheat [41]. GRM was imported from China, and it has dimensions similar to those of a power tiller. It is manually operated, with the operator walking behind the machine.

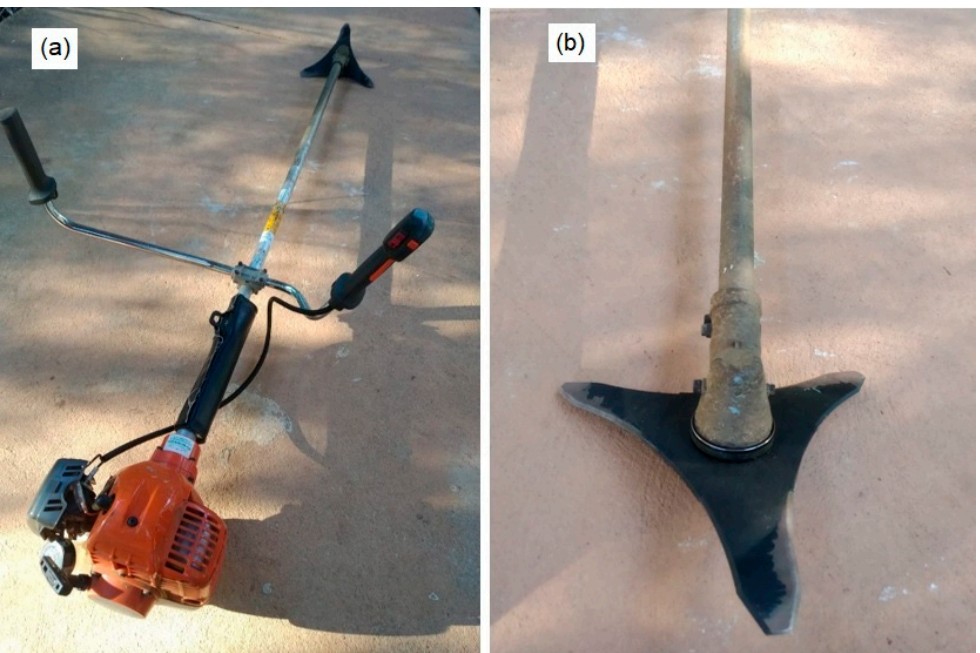

**Figure 4.** (**a**) Costal brushcutter; (**b**) rotating blades.

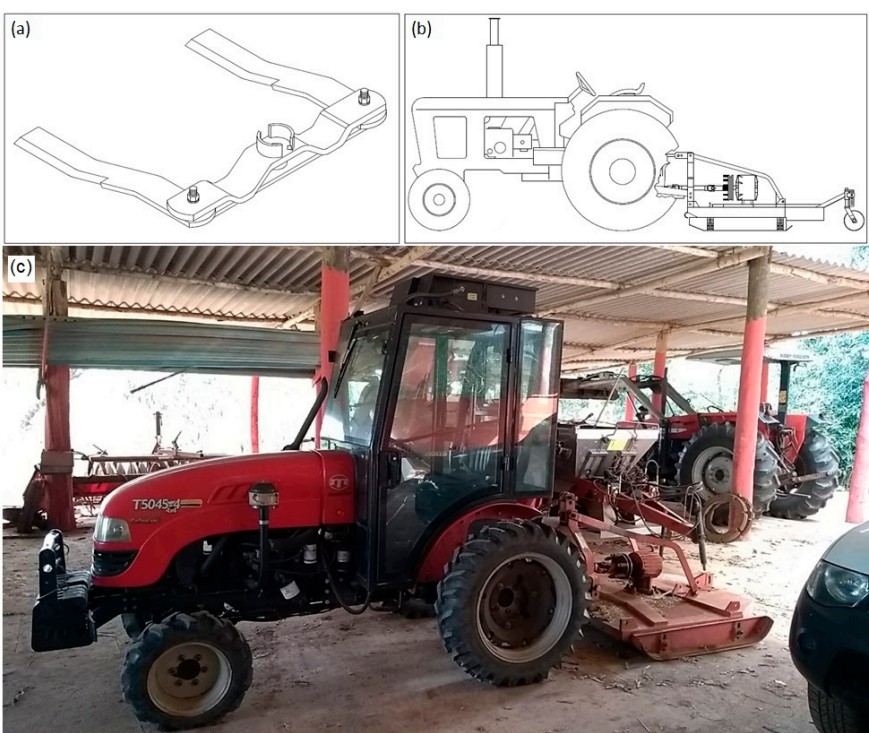

**Figure 5.** Tractor mounted rotary brushcutter: (**a**) rotating blades; (**b**) implement connected to the tractor's three-point system; (**c**) tractor and implement utilized.

The self-propelled Reaper Machine model 4G–120A (Figure 6) has a working width of 1.2 m and a minimum cutting height of 5 cm. Its dimensions are 2.05 m in length and 0.57 m in height. It can operate at a speed ranging from 2.6 to 6.2 km/h. With a weight of up to 120 kg, it utilizes a gear transmission and is equipped with an engine power of 6.6 kw (9 CV). Its fuel consumption is estimated to be between 8.15 and 12.11 L per hectare. The machine's cutting system features a divider assembly that separates and guides the material towards the cutting blades. Subsequently, the biomass is pushed to the right by the conveyor chain and the star wheels, and it is deposited on the ground after being cut.

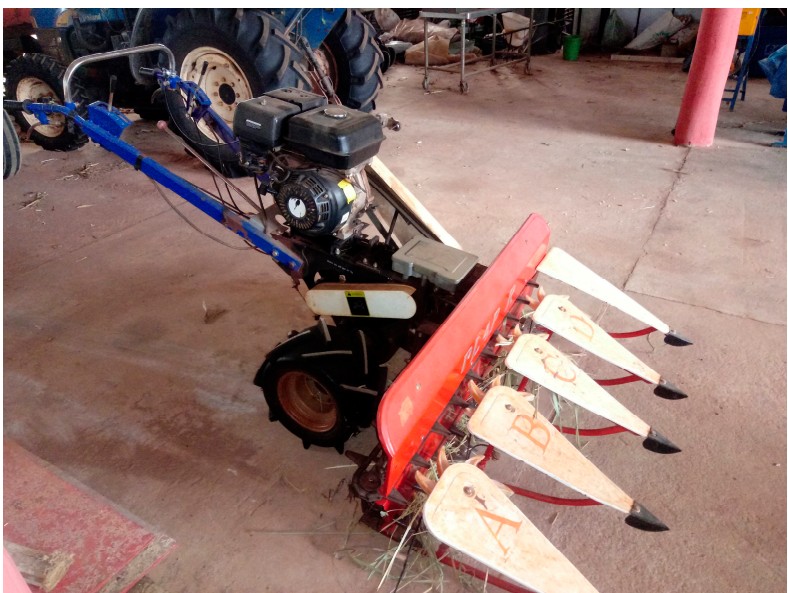

**Figure 6.** Mini grain reaper machine.

*2.5. Methods*

2.5.1. Experimental Design

The experimental design used in this study (Figure 7) was based on randomized blocks, as described by [34]. The project was conducted in a completely randomized manner and consisted of six blocks, with each block divided into three plots, corresponding to a different treatment. The grass cutting was performed in three cycles over the course of one year, and data were collected after each cycle. Each block was 49 m long and 6 m wide, for a total area of 294 m², while each plot measured 13 m long by 6 m wide, equaling 78 m². Additionally, there were two areas for maneuvering the machines between the plots, measuring 5 m long and 6 m wide each, for a total of 30 m². The grass planting area for the experiment was 1.404 m², with the total area of the blocks being 1.764 m².

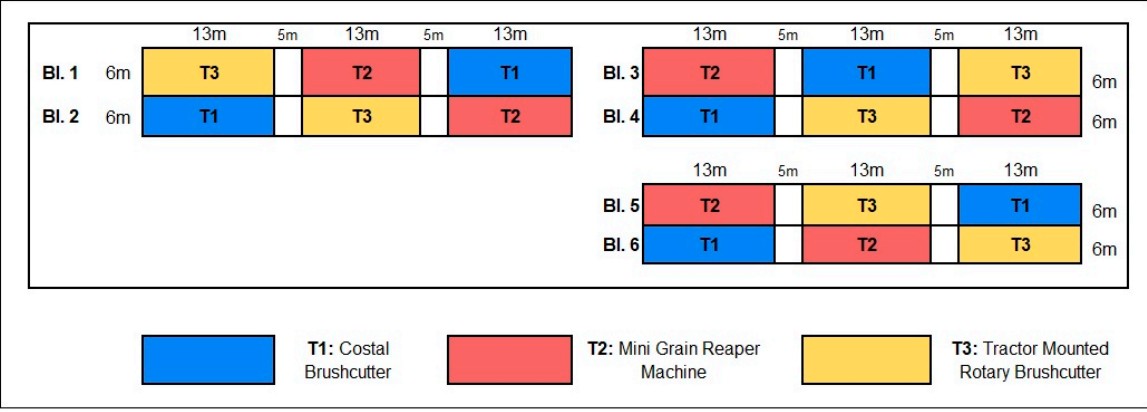

**Figure 7.** Experimental design.

After each grass cutting cycle for the various treatments, three randomly selected sample units measuring 50 cm by 50 cm (0.25 m²) were established within each plot. Within these sample units, data were collected to assess the post-cut grass growth, including regrowth speed (RS), cutting height (CH), and the count of the number of basal tillers and regrowth emissions. The dry mass (DM) samples were obtained from 1 m² sample units, with the vertices aligned with the previously mentioned sample units.

2.5.2. Data Collection

The study centered on the mechanized grass-cutting process and included measuring specific parameters at regular intervals throughout the experiment. The grass cutting was conducted in three cycles corresponding to different treatments: from April to November 2021 (fall, winter, and spring), from November 2021 to January 2022 (spring and summer), and from January to April 2022 (summer and fall). A cutting cycle represents the time interval between two consecutive grass cuttings, and its duration varies based on the grass growth rates, with longer cycles occurring during fall and winter and shorter cycles occurring during spring and summer due to fluctuating plant growth rates throughout the year.

- Count of basal tiller and regrowth. This was conducted to assess grass regrowth speed (RS). This assessment involved tallying the total number of tillers and regrowth within the sample units [42], 21 days after each grass cutting cycle. This specific timeframe was chosen as it enabled the accurate identification of the tillers and regrowths in the tussocks.
- Regrowth speed (RS). This was assessed by measuring the length of leaves in selected regrowths within the sampling units following each treatment cycle. The length was measured twice, at 7 and 42 days after cutting, resulting in a total of 12 measurements per plot and 4 per sampling unit. Both tillers and regrowths were included in the measurements, with 2 measurements for each. To calculate the RS in centimeters per

day, the difference between the final and initial length values was divided by the number of days involved, according to Equation (2), as suggested by [43]:

$$RS = (FL - IL)/ND \tag{2}$$

where RS = grass regrowth speed (cm/day); FL = grass final length (cm); IL = grass initial length (cm); and ND = number of days involved (days).

Subsequently, the twelve values obtained in each plot were averaged to derive a single RS value. This calculation took into consideration the number of tillers and regrowths present in the sample units. Initially, the average RS values for tillers (tRSa) and regrowths (rRSa) were computed. The percentages of tillers and regrowths were determined by counting their numbers within each sampling unit. The RS averages per sampling unit were then calculated using Equation (3). Finally, the RS value for the plot was obtained by averaging the values of the sampling units:

$$RS(su) = (tRSa \times tP) + (rRSa \times rP) \tag{3}$$

where RS(su) = grass RS per sampling unit (cm/day); tRSa = tiller RS average (cm/day); tP = tiller percentage in the sampling unit (%); rRSa = regrowth RS average (cm/day); and rP = regrowth percentage in the sampling unit (%).

- Cutting height (CH). This was determined by positioning a ruler vertically adjacent to the tussocks and measuring the distance from the ground to the point of the cut made by the machines [44,45]. In order to obtain the plot value, nine measurements were randomly taken within the sample units, and the simple average was computed.
- Handling time (HT). This refers to the duration taken by the machines to cut the grass within each plot, which was meticulously recorded. The grass in each plot was systematically divided into strips, and the cutting time for each individual strip was cumulatively summed to derive the total HT for each treatment. It is important to note that the time allocated for maneuvering the machines was not taken into account when calculating the HT.
- Fuel consumption value (FC). This was calculated to compare the fuel consumption efficiency of machines with different characteristics, including size, weight, engine, and fuel type. The correlation analysis considered three key factors: (a) fuel price, (b) handling time per treatment (HT), and (c) fuel consumption of each machine. The methodology used to determine these values is elucidated below:

  (a) Fuel price: The fuel price values in "reais" (R$) per liter were obtained using publicly available data from the National Petroleum, Natural Gas, and Biofuels Agency [46], which is linked to the Ministry of Mines and Energy (MME) of the Federal Government of Brazil. The data were obtained through the agency's Price Survey System (SLP), and they were filtered based on the types of fuel used (gasoline and diesel), the location of the fuel purchase (Campinas, São Paulo), and the specific one-week period corresponding to the days on which the grass was cut.

  (b) Handling time (HT): The HT values were determined as described earlier in the methodology section.

  (c) Fuel consumption: This was determined based on the size and handling capacity of the machines, using two distinct methods.

- For the CBC and GRM, both of which are small and utilize gasoline as fuel, the fuel tank was emptied and a precisely measured amount of gasoline was added. The machines were then operated under working conditions, cutting Mombaça grass, until the entire quantity of gasoline was fully consumed. The FC values were calculated, as suggested by [45], by dividing the volume of gasoline consumed (in liters) by the total working time (in hours).

- In order to determine the FC value of the tractor that was coupled to the RBC, a different methodology was employed. As the tractor is a large machine powered by diesel fuel, it was not practical to measure the fuel consumption directly. Therefore, the specific method in [47] was utilized as a reference to calculate the machine fuel consumption (Equation (4)):

$$mfc = SFC \times Y \tag{4}$$

  where mfc = machine fuel consumption (L/h); SFC = specific fuel consumption (L/kW.h); and Y = current power delivery (KW).

- Using the obtained values of "a", "b", and "c", a correlation was established among them (Equation (5)) in order to calculate the FC in basic dimensions:

$$FC = HT \times mfc \times FP \tag{5}$$

  where FC = fuel consumption value (R$/ha); HT = handling time in each plot (h/ha); mfc = machine fuel consumption (L/h); and FP = fuel price (R$/L).

- Labor cost (LC). The machines employed for grass cutting in each plot have varying operation times and the operators of these machines also have different wages. To facilitate the comparison of LC across treatments, it was necessary to establish a correlation between (a) labor cost and (b) handling time (HT) in order to ensure the compatibility of the values. The specific methodology employed to calculate these factors is elaborated in the following sections:

  (a) Labor value: Two roles were taken into account for machine operation: tractor operator for the tractor and monthly employee for CBC and GRM. The hourly wages for these roles were sourced from publicly available data provided by the Institute of Agricultural Economics [48], an organization associated with the State of São Paulo Government's Secretary of Agriculture and Supply. The data was filtered based on the roles, the region (state of São Paulo), and the grass-cutting dates to determine the respective hourly wage rates (in R$/month) for each role.

  (b) Handling time (HT): HT values were utilized in accordance with the aforementioned description.

- Once the values "a" and "b" were obtained, a correlation was established between them (Equation (6)) in order to calculate the LC in basic dimensions:

$$LC = HT \times LV \tag{6}$$

  where LC = labor cost (R$/ha); HT = handling time (hour/ha); and LV = labor value (R$/hour).

- Dry matter weight (DM). This was assessed through manual biomass collections carried out prior to each grass cutting using pruning shears. The collections were randomly conducted within a 1 m$^2$ template, keeping a distance of 10 cm from the ground. The collected materials were carefully placed in sealed plastic bags, were weighed immediately to obtain the fresh weight, and were then transferred to paper bags for drying in an oven that was set at 50 °C until a stabilized weight was achieved [44,49]. The weight of the dried mass was recorded. The DM value was calculated by multiplying the weight of the green mass of the corresponding sample by the percentage of dry mass of each subsample.

- Raking time of cut grass (RT). This was determined by measuring the duration of the process involved in gathering and arranging the cut grass into windrows. This step aimed to simulate the handling of the grass within the AFS rows. The collected biomass was divided into two equal parts, with one half raked and piled up on one edge of the plot and the other half piled up on the opposite edge. The time taken to complete this task carefully was recorded.

### 2.5.3. Statistical Analysis

The Descriptive Statistics used in this study include several key components: the number of samples, averages, variance, standard deviation, coefficient of variation, range, skewness, and kurtosis. The statistical analysis was conducted using R Studio software. All treatment data underwent tests for kurtosis (k < 3 and k > −3) and symmetry (g < 3 and g > −3), as recommended by [50].

The Exponentially Weighted Moving Average (EWMA) was utilized as a statistical control method, as recommended by [33], as it is well suited for situations where the data do not exhibit normality. This method was originally proposed by [51]. According to [52], EWMA control charts are frequently employed to identify slight fluctuations in data patterns, offering an estimation of the new process mean that may impact the desired quality characteristic. As mentioned by [33], EWMA control charts are particularly suitable for individual observations, as this statistical method calculates the EWMA value as a weighted moving average with geometric progression weights.

The construction of the EWMA control chart is constructed by plotting "$Zi$" against the sample number "$i$" [52] on a chart that includes a center line at $\mu_0$ and appropriate control limits [33]. The chart is defined by Equation (7):

$$Z_i = \lambda x_i + (1 - \lambda)Z_i - 1 \tag{7}$$

where $0 < \lambda \leq 1$; $Z_i = \mu_0 = \bar{x}$ (target value or mean value in control of $x_i$).

The variance of the variable "$z$" is obtained by Equation (8):

$$\sigma^2\_(Z_i) = \sigma^2 \times [\lambda/(2 - \lambda)] \times [1 - (1 - \lambda)^{2i}] \tag{8}$$

where $\sigma$ = standard deviation of the data in relation to the mean; n = size of samples; $\lambda$ = weight given to each sample; and $i$ = sample order used.

The upper control limits (UCL) and lower control limits (LCL) for the EWMA control charts are obtained according to Equations (9) and (10), respectively:

$$UCL = \bar{x} + L\sigma \sqrt{\left\{[\lambda/(2 - \lambda)] \times \left[1 - (1 - \lambda)^{2i}\right]\right\}} \tag{9}$$

$$LCL = \bar{x} - L\sigma \sqrt{\left\{[\lambda/(2 - \lambda)] \times \left[1 - (1 - \lambda)^{2i}\right]\right\}} \tag{10}$$

where UCL = upper control limits; LCL = lower control limits; $\bar{x}$ = data mean; $\sigma$ = standard deviation of the data in relation to the mean; $\lambda$ = weight given to each sample; and $i$ = sample order used. Central Line = $\mu_0$ = x.

### 2.5.4. Calculation of Minimum Number of Samples

For calculating the minimum number of samples required to be collected each plot, it was necessary to conduct a preliminary survey of the data that are related to the parameter being calculated. This would allow the number of samples needed to achieve normality in the experiment to be determined through the standard mean error. The minimum number of samples can be determined based on the means and the standard deviations of the preliminary data as well as by similar studies found in the literature.

Standard mean error (Equation (11)) and standard deviation (Equation (12)) were calculated according to [53] as follows:

$$d = (|\mu - \mu_0|)/\sigma \tag{11}$$

where d = standard mean error; $\mu$ = estimated average; $\mu_0$ = average of the data collected; and $\sigma$ = standard deviation.

$$\sigma = [\Sigma (x_i - \mu_0)^2]/n \tag{12}$$

where σ = standard deviation; $x_i$ = obtained data value; $\mu_0$ = average of the data collected; and n = amount of data collected.

Standard mean error was then plotted on a chart of operating characteristic curves (Figure 8) with a significance level α = 0.05 where the "x" axis represents the value of the standard mean error found in Equation (11), while the "y" axis represents the beta error β, which is the maximum tolerable error in the experiment. The curves "n" refer to the minimum number of samples [53].

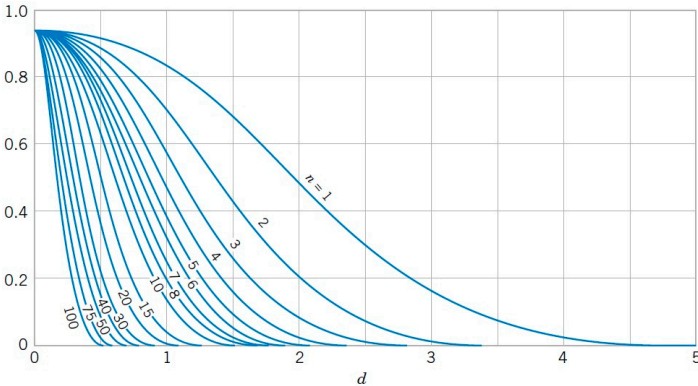

**Figure 8.** Chart of operating characteristics curves.

## 3. Results

Three different machines were assessed for cutting Mombaça grass biomass in this investigation: the mini grain reaper machine (GRM), the costal brushcutter (CBC), and the tractor mounted rotary brushcutter (RBC). Each machine executed the same task in its unique manner, and throughout the three cutting cycles of the grass plots, seven parameters were measured and analyzed in order to observe and delineate the characteristics of the cutting methods employed by the machines.

In relation to the intrinsic characteristics of the machines, the following parameters were observed: the machine handling time (HT—h/ha), the fuel consumption value (FC—R\$/ha), the labor cost (LC—R\$/ha) of the operator, and the grass-raking time (RT—h/ha) required to both gather and stack the entire harvested biomass. The response of Mombaça grass in terms of regrowth after each machine cut was also assessed and evaluated based on the following parameters: cutting height (CH—cm), grass regrowth speed (RS—mm/day), and dry mass production (DM—ton/ha).

### 3.1. Statistical Analyses

The experiment comprised three cutting cycles of Mombaça grass under the different treatments. Thus, the descriptive statistics table below is provided for each cutting cycle, displaying the results of each treatment (Tables 1–3).

**Table 1.** Descriptive statistics for Cycle 1 (CH = cutting height; RS = regrowth speed; DM = dry mass; HT = handling time; FC = fuel consumption; LC = labor cost; RT = raking time).

| | | CH | RS | DM | HT | FC | LC | RT |
|---|---|---|---|---|---|---|---|---|
| | | (cm) | (mm/Day) | (ton/ha) | (h/ha) | (R\$/ha) | (R\$/ha) | (h/ha) |
| | Samples | 18 | 18 | 18 | 6 | 6 | 6 | 6 |
| | Average | 8.89 | 6.07 | 2.52 | 5.82 | 41.59 | 36.75 | 27.78 |
| | Variance | 4.22 | 2.02 | 0.51 | 5.40 | 276.01 | 215.47 | 40.12 |
| Mini Grain Reaper Machine | Standard Deviation | 2.05 | 1.42 | 0.71 | 2.32 | 16.61 | 14.68 | 6.33 |
| | Coefficient of Variation | 23.12 | 23.40 | 28.24 | 39.94 | 39.94 | 39.94 | 22.81 |
| | Range | 6.00 | 6.90 | 2.77 | 5.38 | 38.46 | 33.98 | 13.89 |
| | Kurtosis | 1.74 | 6.32 | 3.34 | 1.55 | 1.55 | 1.55 | 1.30 |
| | Skewness | 0.20 | −1.26 | 0.70 | 0.62 | 0.62 | 0.62 | −0.22 |

**Table 1.** *Cont.*

|  |  | CH | RS | DM | HT | FC | LC | RT |
|---|---|---|---|---|---|---|---|---|
|  |  | (cm) | (mm/Day) | (ton/ha) | (h/ha) | (R$/ha) | (R$/ha) | (h/ha) |
| Costal Brushcutter | Samples | 18 | 18 | 18 | 6 | 6 | 6 | 6 |
|  | Average | 12.22 | 5.82 | 2.50 | 34.22 | 208.38 | 216.18 | 37.50 |
|  | Variance | 5.95 | 2.59 | 0.35 | 1.28 | 47.70 | 51.28 | 30.10 |
|  | Standard Deviation | 2.44 | 1.61 | 0.59 | 1.13 | 6.91 | 7.16 | 5.49 |
|  | Coefficient of Variation | 19.95 | 27.61 | 23.63 | 3.31 | 3.31 | 3.31 | 14.63 |
|  | Range | 10.00 | 6.55 | 2.40 | 3.20 | 19.52 | 20.24 | 13.89 |
|  | Kurtosis | 3.33 | 3.03 | 2.99 | 3.13 | 3.13 | 3.14 | 2.61 |
|  | Skewness | 0.24 | 0.51 | 0.45 | 1.08 | 1.08 | 1.09 | 0.94 |
| Tractor Mounted Rotary Brushcutter | Samples | 18 | 18 | 18 | 6 | 6 | 6 | 6 |
|  | Average | 8.61 | 5.08 | 2.13 | 5.50 | 160.29 | 46.72 | 36.11 |
|  | Variance | 2.84 | 3.88 | 0.33 | 0.05 | 41.66 | 3.54 | 92.55 |
|  | Standard Deviation | 1.69 | 1.97 | 0.57 | 0.22 | 6.45 | 1.88 | 9.62 |
|  | Coefficient of Variation | 19.57 | 38.83 | 26.94 | 4.07 | 4.03 | 4.03 | 26.64 |
|  | Range | 6.00 | 7.24 | 2.23 | 0.61 | 17.65 | 5.15 | 19.44 |
|  | Kurtosis | 2.84 | 2.46 | 3.61 | 2.41 | 2.43 | 2.43 | 1.26 |
|  | Skewness | 0.79 | 0.35 | 1.04 | 0.83 | 0.83 | 0.83 | −0.25 |

**Table 2.** Descriptive statistics for Cycle 2 (CH = cutting height; RS = regrowth speed; DM = dry mass; HT = handling time; FC = fuel consumption; LC = labor cost; RT = raking time).

|  |  | CH | RS | DM | HT | FC | LC | RT |
|---|---|---|---|---|---|---|---|---|
|  |  | (cm) | (mm/Day) | (ton/ha) | (h/ha) | (R$/ha) | (R$/ha) | (h/ha) |
| Mini Grain Reaper Machine | Samples | 18 | 18 | 18 | 6 | 6 | 6 | 6 |
|  | Average | 7.65 | 12.18 | 3.76 | 5.59 | 48.32 | 36.84 | 30.56 |
|  | Variance | 6.27 | 16.98 | 3.13 | 0.52 | 38.91 | 22.60 | 46.30 |
|  | Standard Deviation | 2.50 | 4.12 | 1.77 | 0.72 | 6.24 | 4.75 | 6.80 |
|  | Coefficient of Variation | 32.73 | 33.84 | 47.05 | 12.89 | 12.91 | 12.90 | 22.27 |
|  | Range | 9.50 | 15.32 | 7.33 | 1.93 | 16.69 | 12.72 | 16.67 |
|  | Kurtosis | 2.95 | 2.74 | 4.38 | 1.78 | 1.78 | 1.78 | 2.04 |
|  | Skewness | 0.54 | 0.53 | 1.24 | −0.15 | −0.15 | −0.15 | 0.63 |
| Costal Brushcutter | Samples | 18 | 18 | 18 | 6 | 6 | 6 | 6 |
|  | Average | 11.18 | 14.14 | 3.65 | 10.83 | 79.65 | 71.31 | 33.33 |
|  | Variance | 8.76 | 25.17 | 1.35 | 2.55 | 138.05 | 110.68 | 27.79 |
|  | Standard Deviation | 2.96 | 5.02 | 1.16 | 1.60 | 11.75 | 10.52 | 5.27 |
|  | Coefficient of Variation | 26.47 | 35.49 | 31.85 | 14.75 | 14.75 | 14.75 | 15.82 |
|  | Range | 11.50 | 20.66 | 4.45 | 4.52 | 33.27 | 29.79 | 16.67 |
|  | Kurtosis | 3.23 | 3.33 | 2.74 | 3.72 | 3.71 | 3.71 | 3.00 |
|  | Skewness | 0.82 | 0.51 | 0.59 | 1.47 | 1.47 | 1.47 | 0.00 |
| Tractor Mounted Rotary Brushcutter | Samples | 18 | 18 | 18 | 6 | 6 | 6 | 6 |
|  | Average | 7.81 | 10.37 | 2.90 | 4.62 | 172.77 | 40.57 | 43.06 |
|  | Variance | 3.83 | 13.70 | 0.95 | 0.03 | 40.54 | 2.24 | 113.44 |
|  | Standard Deviation | 1.96 | 3.70 | 0.98 | 0.17 | 6.37 | 1.50 | 10.65 |
|  | Coefficient of Variation | 25.09 | 35.68 | 33.59 | 3.68 | 3.69 | 3.69 | 24.74 |
|  | Range | 8.00 | 12.17 | 3.44 | 0.44 | 16.45 | 3.87 | 30.56 |
|  | Kurtosis | 3.40 | 1.96 | 3.05 | 1.70 | 1.70 | 1.70 | 3.69 |
|  | Skewness | −0.35 | 0.18 | 1.01 | 0.15 | 0.14 | 0.14 | 1.45 |

**Table 3.** Descriptive statistics for Cycle 3 (CH = cutting height; RS = regrowth speed; DM = dry mass; HT = handling time; FC = fuel consumption; LC = labor cost; RT = raking time).

|  |  | CH | RS | DM | HT | FC | LC | RT |
|---|---|---|---|---|---|---|---|---|
|  |  | (cm) | (mm/Day) | (ton/ha) | (h/ha) | (R$/ha) | (R$/ha) | (h/ha) |
| Mini Grain Reaper Machine | Samples | 18 | 18 | 18 | 6 | 6 | 6 | 6 |
|  | Average | 7.81 | 12.57 | 6.80 | 10.51 | 89.72 | 73.55 | 34.72 |
|  | Variance | 5.26 | 20.89 | 7.26 | 2.69 | 196.14 | 131.77 | 113.45 |
|  | Standard Deviation | 2.29 | 4.57 | 2.69 | 1.64 | 14.01 | 11.48 | 10.65 |
|  | Coefficient of Variation | 29.39 | 36.36 | 39.62 | 15.61 | 15.61 | 15.61 | 30.67 |
|  | Range | 9.25 | 16.21 | 9.31 | 4.63 | 39.52 | 32.39 | 30.56 |
|  | Kurtosis | 3.67 | 2.14 | 2.77 | 2.58 | 2.57 | 2.57 | 3.69 |
|  | Skewness | 0.87 | 0.14 | 0.97 | −0.70 | −0.69 | −0.69 | 1.45 |
| Costal Brushcutter | Samples | 18 | 18 | 18 | 6 | 6 | 6 | 6 |
|  | Average | 10.40 | 13.07 | 6.83 | 11.28 | 82.01 | 78.95 | 42.13 |
|  | Variance | 17.82 | 21.31 | 7.96 | 2.27 | 119.95 | 111.11 | 492.04 |
|  | Standard Deviation | 4.22 | 4.62 | 2.82 | 1.51 | 10.95 | 10.54 | 22.18 |
|  | Coefficient of Variation | 40.58 | 35.32 | 41.33 | 13.35 | 13.35 | 13.35 | 52.65 |
|  | Range | 14.25 | 17.02 | 10.53 | 4.27 | 31.07 | 29.90 | 61.11 |
|  | Kurtosis | 3.06 | 3.69 | 5.03 | 2.90 | 2.90 | 2.90 | 3.81 |
|  | Skewness | 1.01 | 1.27 | 1.64 | −1.04 | −1.05 | −1.05 | 1.57 |
| Tractor Mounted Rotary Brushcutter | Samples | 18 | 18 | 18 | 6 | 6 | 6 | 6 |
|  | Average | 6.49 | 11.93 | 5.67 | 5.74 | 224.06 | 53.26 | 46.30 |
|  | Variance | 6.70 | 28.54 | 10.89 | 0.20 | 312.78 | 17.67 | 26.74 |
|  | Standard Deviation | 2.59 | 5.34 | 3.30 | 0.45 | 17.69 | 4.20 | 5.17 |
|  | Coefficient of Variation | 39.91 | 44.80 | 58.23 | 7.88 | 7.89 | 7.89 | 11.17 |
|  | Range | 12.00 | 23.05 | 12.62 | 1.12 | 43.67 | 10.38 | 11.11 |
|  | Kurtosis | 7.73 | 5.51 | 4.45 | 1.58 | 1.58 | 1.58 | 1.18 |
|  | Skewness | 1.93 | 1.44 | 1.38 | 0.03 | 0.02 | 0.02 | 0.12 |

As can be observed in the tables, numerous skewness and kurtosis values fell outside the range of −3 and 3, as specified by [50], for normal data distribution, which indicates a non-normal distribution of the data. Hence, in accordance with the suggestion of [33], EWMA plots were employed as a statistical control to compare the variability of the means.

The key parameter selected for comparison using EWMA plots (Figure 9) was the dry matter production (DM). This choice was based on the primary objective of the experiment, which aimed to identify the machine that yields the highest biomass production, reflecting the quality of grass regrowth following successive cuts over a one-year period. While studying sugarcane, [54] stated that the cutting quality is an important indicator of harvester performance because it can reduce productivity losses and maintain high yield in the following harvest.

The symbol "+" means the position on the chart the sample was before the data being submitted to EWMA statistical control method. Based on the definitions provided by [55,56], the process is considered stable when 95% or more of the points fall within the upper and lower control limits (UCL and LCL), characterized as black points. Examining the EWMA charts depicted in Figure 9, it can be observed that GRM consistently exhibited a stable process throughout all three cycles, with the majority of data points falling within the predetermined limits. The CBC also showed a stable process in Cycles 1 and 2, but in Cycle 3, more than 5% of the data points exceeded the specified limits (characterized as red points). Conversely, RBC initially exhibited an unstable process in the first two cycles, with data points surpassing the control limits. Nevertheless, in Cycle 3, the process became stable as the majority of the data points fell within the defined limits.

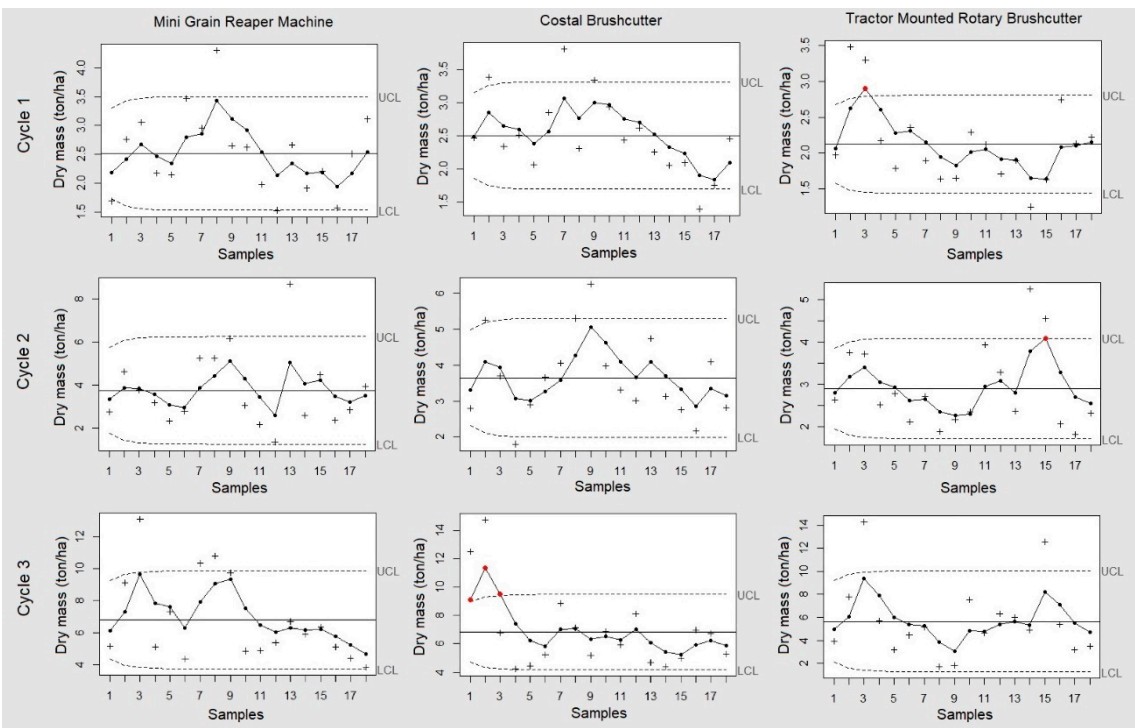

**Figure 9.** EWMA charts for dry mass (DM) for cycles 1, 2 and 3.

Upon analyzing the charts, it was observed that the cuts performed by GRM and CBC, which also exhibited higher values of CH and RS, were associated with the highest DM production in all three cycles. Although the average production values varied in each cycle, the overall average values were comparable between the GRM and the CBC treatments. This indicates that these cuts consistently led to higher DM production throughout the cycles. In contrast, the RBC treatment yielded the lowest DM production in all cycles, indicating that the cuts made by this treatment were not effective in maximizing production. When comparing the DM productions over two years, [57] observed lower DM production of grasses under grazing compared to those manually cutting, possibly because grazing damages the grass more significantly.

Therefore, the results suggest that the cuts performed by GRM and CBC, despite exhibiting higher variability, were more successful in maximizing DM production compared to RBC. Furthermore, maintaining process stability is crucial for ensuring consistent and predictable outcomes. In the case of RBC, the instability observed in the first two cycles may have had a detrimental impact on DM production. However, in the third cycle, with a stable process, the production levels approached those of the other treatments.

### 3.2. Characterization of the Treatments' Cutting Types in Mombaça Grass

Figure 10 illustrates the results observed immediately after the grass was cut by the mini grain reaper machine (GRM) and the piling up of the harvested biomass.

According to Figure 10, the following can be observed: (a) the machine and the grass plot mowed by it, prior to piling up the cut biomass; (b) the grass tussocks cut by this machine, exhibiting superior cutting quality compared to the other machines, as evidenced by minimal damage to the plants and apical buds; (c) the piled-up grass biomass with intact and non-shredded leaves; and (d) the grass plot mowed by this machine, displaying uneven cutting as indicated by the presence of grass strands and several grass tussocks that were not completely cut.

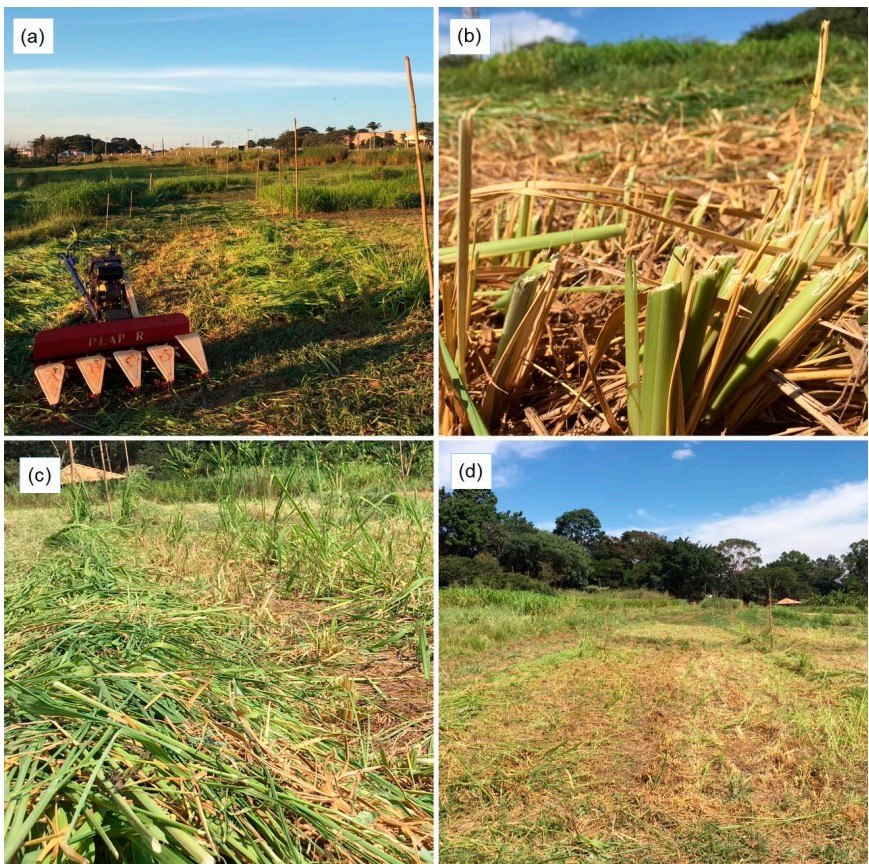

**Figure 10.** GRM's cutting type characterization: (**a**) Mini grain reaper machine; (**b**) the grass tussocks cut by this machine; (**c**) the piled-up grass biomass; (**d**) the grass plot after biomass removal.

The GRM, being a reaper-type harvester, demonstrated superior cutting quality compared to the other machines evaluated. Its cutting process resulted in minimal damage to the grass tussocks and exposed only the internal tissues of the plant in the cutting area, without causing cracks in the remaining grass, as observed in the cutting performed by the brushcutters.

The design of this machine is intended for harvesting crops such as rice and wheat [41], which have upright and fibrous stems and stalks that facilitate the transport of harvested material to the side of the machine. As it moves through the field, the harvested material is then deposited on the right side, preventing clogging in the cutting system. However, when used to harvest the leaves of Mombaça grass, which lack upright and fibrous stems, the machine inevitably experiences clogging. This leads to partially cut grass tussocks and also increases the machine's handling time (HT), as the machine needs frequent stops to unclog. The time required for unclogging was included in the HT, resulting in increased values for other parameters related to HT, such as fuel consumption (FC) and labor cost (LC).

GRM has also been tested in Egypt for bean harvesting [58]. The authors observed that the machine experienced clogging issues, leading to significant yield losses of over 50%. Higher operating speeds further exacerbated the losses due to the excessive plant load on the cutting bar.

Figure 11 illustrates the result of cutting by GRM and how it deposits the cut grass leaves, already piled up, beside the machine in an ideal situation without clogging. This method of biomass deposition, with entire and non-shredded leaves, facilitated the stacking process, resulting in the shortest raking time (RT).

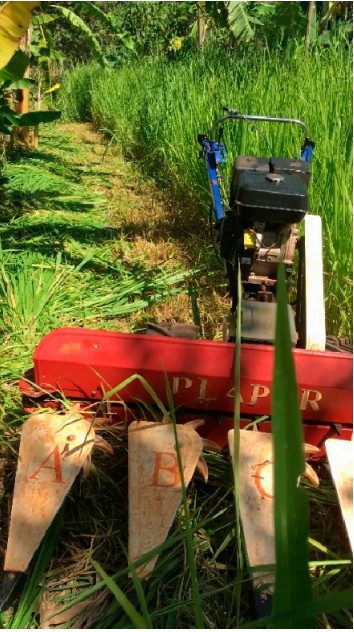

**Figure 11.** Deposition of cut grass leaves by GRM.

The following figure (Figure 12) demonstrates the results obtained after cutting the grass by the costal brushcutter (CBC) and raking the biomass.

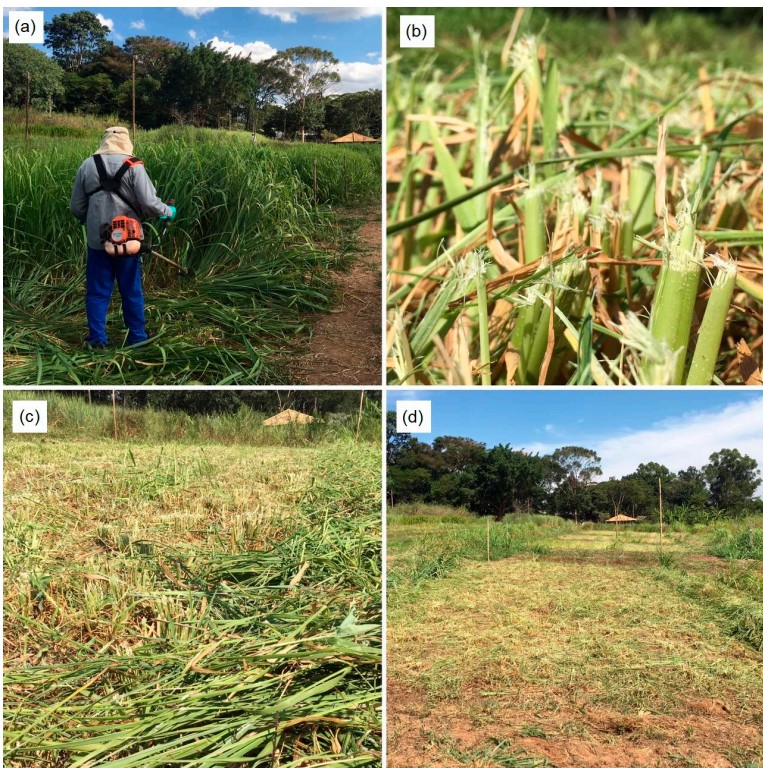

**Figure 12.** CBC's cutting type characterization: (**a**) costal brushcutter and the operator cutting the grass; (**b**) the grass tussocks cut by this machine; (**c**) the piled-up grass biomass; (**d**) the grass plot after biomass removal.

According to Figure 12, the following can be observed: (a) the machine and the grass in the plot being cut, where it is possible to see the harvested biomass being scattered across the plot; (b) the cut grass tussocks, where a medium quality of cutting can be observed

compared to other machines, as evidenced by the damage to the plants only at the cutting area while keeping the apical buds intact; (c) the piled grass biomass with entire and non-shredded leaves; and (d) the plot after piling, where the absence of grass strands results in a homogeneous cut.

Compared to the GRM, the CBC enables a similar regrowth process with new basal tiller emission and regrowth from apical buds, possibly resulting from a cut high enough to keep the apical buds untouched. It was observed that both treatments resulted in more regrowth from apical buds than basal tiller emission. The same results were observed by [42] when comparing the sugarcane regrowth process after being cut by sharp and worn blades, and the authors observed no difference in sugarcane regrowth.

CBC cut the grass 11.27 cm average from the soil, the highest values observed, and [59,60] demonstrated even higher values using a CBC (15 cm and 20 cm, respectively). However, the GRM provides a more precise cut, minimizing damage to grass plants. In contrast, the CBC causes greater plant damage, exposing internal tissues and potentially increasing the risk of contamination. Additionally, water loss is possibly higher when using the CBC, leading to increased stress levels.

It was observed during the experiments that the grass cut by the CBC resulted in discarded and dried-up plant parts, while the grass cut by the GRM retained live plant parts that performed photosynthesis. This difference may lead to increased stress for the grass cut by the CBC compared to the GRM, potentially affecting the regrowth vigor.

The cutting process of the CBC results in whole cut leaves, similar to the cutting process of GRM. Nonetheless, due to the scattered deposition of the cut biomass, the grass cut by this machine required a longer raking time (RT).

Figure 13 shows the results obtained after the grass cutting performed by the tractor mounted rotary brushcutter (RBC) and the biomass raking.

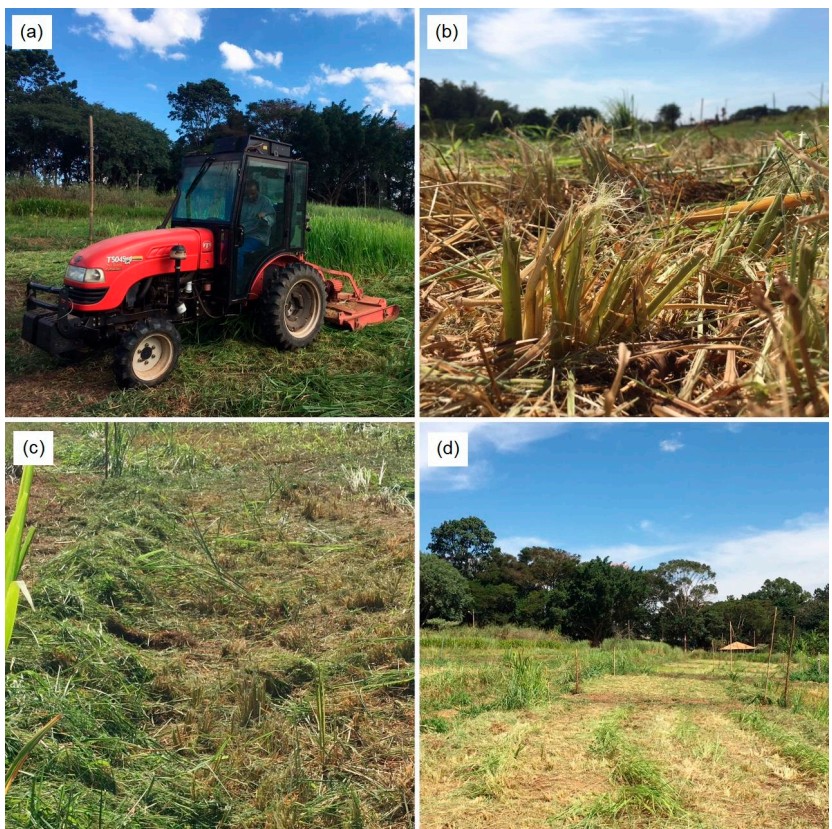

**Figure 13.** RBC's cutting type characterization: (**a**) tractor and rotary brushcutter implement (**b**) the grass tussocks cut by this machine; (**c**) the piled-up grass biomass; (**d**) the grass plot after biomass removal.

According to Figure 13, the following can be observed: (a) tractor and RBC implement; (b) the cut tussock by this machine, where the cut's lower quality can be noticed compared to the other machines, as evidenced by the total damage to the remaining tussock, including the apical buds; (c) the raked biomass of shredded grass leaves; and (d) the plot of grass mowed by RBC, where the presence of grass strands resulting from an uneven cut can be seen, leading to several tussocks of grass not completely cut.

Compared to the other machines, RBC was characterized as the one responsible for causing the most damage to the grass tussocks. Its cutting was performed similarly to CBC, as it uses rotating blades to impact and cut the grass, with the difference being that the RBC blades are thicker than those of CBC. This low cutting quality associated with an unsharpened knife results even in efficiency losses and a higher expenditure of energy [61], fuel, and capital. Impact cutting requires very sharp blades when aiming to minimize damage and reduce the chances of grass contamination by pathogens. The higher number of cracks in the remaining tussocks caused by RBC resulted in a larger area of exposure of the internal plant tissues, possibly leading to a higher risk of contamination and water loss.

RBC cutting was low and harmful enough to damage apical buds, resulting in a higher number of basal tiller emergence during the regrowth process (60%) compared to the other machines (42% and 32% respectively, for GRM and CBC). The number of regrowths from apical buds in response to the RBC type of cut was lower compared to the other cuts.

When comparing the influence of cutting height (CH) on Mombaça grass, [62] observed that the lower the CH, the higher the number of new tillers and the lower the number of regrowths from apical buds emitted during the regrowth process. Indeed, this phenomenon was also observed in this research, as CBC provided the highest CH (11.3 cm average) and percentage of regrowths from apical buds (68%); CH observed for GRM (8.1 cm average) resulted in 58% of regrowths from apical buds; and RBC's CH (7.6 cm average) was responsible for the lowest percentage of regrowths from apical buds (40%).

Furthermore, it is possible that the cutting height and the quality play fundamental roles on dry mass production (DM) as well as on the regrowth speed (RS). Both GRM and CBC, which performed higher cuts and with better quality, were responsible for almost the same weight of DM produced (4.36 and 4.33 ton.ha$^{-1}$ average, respectively), while RBC was responsible for the lowest DM production (3.57 ton.ha$^{-1}$ average). The grass cut by RBC regrew at a rate of 9.1 mm.day$^{-1}$ average, as the grass mowed by GRM and CBC regrew at a rate of, respectively, 10.3 and 11 mm.day$^{-1}$ average.

The previous figure (Figure 13c) demonstrates the result obtained after the cutting by the RBC and the biomass raking. It can be observed that the grass leaves were shredded due to the machine's cutting type. During the experiments, it was observed that the machine not only shredded the leaves but also scattered them across the plot. This fact resulted in the longest raking time (RT) for RBC (42 h.ha$^{-1}$.person$^{-1}$ average) compared to the other machines tested here (38 and 31 h.ha$^{-1}$.person$^{-1}$ average, respectively, for CBC and GRM).

From Figure 13d, it is possible to observe a similar effect on the uneven grass cutting caused by the GRM. In this specific RBC case, tussocks were not completely cut because the cutting mechanism is located at the rear and the tractor moves through the cultivation area, trampling the grass and bending it to the ground. When the implement performs the cutting, even though its cutting height is the lowest compared to other machines, it is not low enough to cut the grass that has been flattened by the tractor. In addition, [61] observed that when rotary blades are utilized, the vegetation cutting process often leads to incomplete cuts as they primarily result in smoothing the vegetation rather than completely severing it. This phenomenon can lead to a lower biomass harvest, lower regrowth vigor, and, consequently, production loss for the next cycle.

### 3.3. Machines Adequacy for Mombaça Grass Biomass Handling in AFS

Table 4 was made to facilitate understanding of the obtained results. The treatment results were listed as averages for the three cutting cycles so that they can be compared to each other. The results were observed by the evaluated parameter values.

**Table 4.** Parameters' averages for the three cutting cycles.

| Treatments | Grass | | | Working Time | | Costs | |
|---|---|---|---|---|---|---|---|
| | CH (cm) | RS (mm/Day) | DM (ton/ha) | HT (h/ha) | RT (h/ha) | FC (R$/ha) | LC (R$/ha) |
| Mini Grain Reaper Machine | 8.12 | 10.27 | 4.36 | 7.31 | 31.02 | 59.88 | 49.05 |
| Costal Brushcutter | 11.27 | 11.01 | 4.33 | 18.78 | 37.65 | 123.34 | 122.15 |
| T. Mounted Rotary Brushcutter | 7.63 | 9.13 | 3.57 | 5.28 | 41.82 | 185.71 | 46.85 |

Regarding the parameters observed in the plant, GRM and RBC presented similar CH homogeneity due to being self-propelled machines, in which the CH is previously set and the machine tends to maintain it, and CBC's CH varied most. RBC presented CH values low enough to damage the plant apical meristem, a fact that resulted in the grass regrowth almost exclusively by the emission of new basal tillers. In accordance with [63], the CH plays a crucial role in determining both the quantity and the quality of the biomass yield.

The CBC cutting system was responsible for the best grass RS results, and both CBC and GRM cutting systems resulted in similar and more superior DM productions than RBC. The DM production is one of the most relevant parameters to indicate the machine with the best cutting system that, due to the successive grass cuts, makes the plant respond with greater biomass production.

Regarding the working time parameters, RBC showed the best HT results, as it was the fastest machine. GRM presented the best RT results due to the integrity of the grass leaves and the way they were deposited on the ground–in bundles –which resulted in less time to rake the grass.

Regarding the financial cost parameters, GRM requires less investment for FC and RBC demands less capital for LC. Although RBC's operator is more expensive than GRM's operator, and RBC's HT was short enough to compensate the costs with personnel. When both financial cost parameters are summed up, as only one machine is to be chosen, GRM was responsible for the best results observed by the lowest cost (108.9 R$.ha$^{-1}$). RBC was responsible for the highest costs with fuel, a fact also observed by [64], who stated that the reduction of power consumption and the increase in efficiency of RBCs is an emergent issue.

The best results provided by the treatments are shown below:

- GRM: dry matter (DM), fuel consumption (FC) and raking time (RT);
- CBC: regrowth speed (RS);
- RBC: handling time (HT) and labor cost (LC).

To complete the machines adequacy analysis and make evident the differences observed in the study, a SWOT (Strengths, Weaknesses, Opportunities, and Threats) analysis (Table 5), one of the oldest and most adopted strategy tools worldwide [65], has been constructed.

**Table 5.** SWOT analysis for the tested machines.

| | Strengths | Weaknesses | Opportunities | Threats |
|---|---|---|---|---|
| Mini Grain Reaper Machine | Superior cutting quality with minimal damage to grass tussocks and apical buds | Clogging issues when used to harvest Mombaça grass | Technological improvements to address clogging issues and optimize performance | Competition from alternative machines with higher capacity and versatility |
| | Provides greater biomass productivity compared to other machines | Increased handling time due to necessary stops for unclogging | Adaptation for grass species or crops with similar characteristics | Potential high costs associated with advanced technological features |

**Table 5.** *Cont.*

| | Strengths | Weaknesses | Opportunities | Threats |
|---|---|---|---|---|
| **Mini Grain Reaper Machine** | Results in whole cut leaves | Uneven grass cutting | Growing demand for high-quality cutting and biomass harvesting machines | Competition with machines commonly available on the market |
| | Deposits the cut grass leaves already piled up | Operational limitations in specific agricultural environments, terrains and grass species | Can be improved with technological advancements to reduce physical effort and increase efficiency | Market unavailability |
| | Provides fastest raking time | - | - | - |
| | Suitable for minimizing operational costs | - | - | - |
| **Costal Brushcutter** | Affordable, cost-effective and easy handling option for small-scale farmers | Slow operation | Can be improved with technological advancements to reduce physical effort and increase efficiency | Competition from other machines with higher efficiency and lower physical effort requirements |
| | Suitable for smaller areas and operations | Higher labor costs | Market demand for affordable and small-scale machinery in agroforestry systems | Potential challenges in adapting the machine for different grass species or agricultural contexts |
| | Provides the fastest regrowth speed | Requires significant labor and physical effort | Reduction in fuel costs by improving machine performance | - |
| | Results in whole-cut leaves | Unsuitable for larger scales | - | - |
| | Do not damage apical buds | - | - | - |
| | Good grass biomass production over time | - | - | - |
| | Homogeneous cutting | - | - | - |
| **Tractor Mounted Rotary Brushcutter** | Suitable for larger-scale contexts and operations | Uneven grass cutting | Potential integration with precision agriculture technologies for enhanced efficiency | Competition from other machines with better stability and lower costs |
| | Reduces working time, physical effort, and labor costs | Higher fuel consumption compared to other machines | Demand for larger-scale machinery in agroforestry systems and commercial agriculture | Potential limitations in maneuverability and accessibility in certain terrains or areas |
| | - | May not be cost-effective for small-scale farmers or limited areas | Potential for modifications to minimize damage to grass plants and improve regrowth vigor | Market saturation and price competition with alternative machines |
| | - | Damages apical buds and tussocks, resulting in productivity losses | Tractor purchase, usage and maintenance can be shared with many farmers | Less availability of skilled labor |
| | - | Grass leaves shredded and scattered after cutting, resulting in longer raking time | Adaptation of the brushcutter to the front of the tractor | - |

It can be concluded that GRM is the most adequate machine to guarantee the best plant response over the successive cuts, which will result in greater biomass production with the lowest operational costs (fuel and labor). It is evident that GRM is the most adequate machine for faster raking the cut grass, or when the availability of labor for this job is low. GRM is especially suggested to farmers who are able to make adaptations, given the excessive obstruction presented in its cutting system during the experiment. Some authors have shown it is possible to make successful adaptations on harvesting machines in order to achieve better results, as it have been done for kenaf and rice harvesting [45,66]. GRM presented superior cutting quality but its functionality made working time less satisfactory.

CBC is especially adequate for smaller areas because it works slowly, requires much physical effort and working time, and is the cheapest machine to purchase and maintain.

It has been demonstrated that RBC is the most adequate machine to carry out the work faster and, therefore, for handling work on larger scales. It is the treatment that allows the least physical effort and costs with personnel when compared to others. This contributes to farmers achieving a better quality of life [67]. RBC is also indicated to farmers who already have a tractor available to work with. Nevertheless, according to [68], mechanized systems with large machines present greater soil compaction.

## 4. Discussion

### 4.1. Practical Application

The key to promoting rapid growth of mechanization in small-scale agriculture lies in the adoption of an appropriate mechanization theory [69]. The study's recommendations on machinery selection can guide farmers and practitioners in choosing the appropriate equipment for agroforestry tasks. This enables them to enhance productivity, reduce physical effort, and minimize operational costs by utilizing machinery that aligns with their specific requirements. Agricultural mechanization has the main objective of rationalizing the use of machinery through applied study. Implementing technology in forestry operations can lead to significant reductions in time costs while enhancing operational efficiency [61].

Thus, it is important to understand that there are principles and processes for selecting machinery, which necessitates planning mechanized systems. When choosing the machines to be used in farming operations, it is important to consider their operational and production characteristics as well as the operating environment, which serves as a mediator between them [70]. Therefore, the adoption of suitable machinery is highly relevant to productivity.

Small-scale agriculture in developing countries has not fully harnessed the untapped potential of agricultural mechanization due to the misconception that it is deemed unworthy for such contexts [69]. The research emphasizes the importance of mechanization for helping small-scale farmers expand the implementation of agroforestry systems to larger areas. By incorporating the recommended machines, farmers and landowners can overcome the limitations of manual labor, facilitate larger-scale adoption of agroforestry practices, and contribute to sustainable land management and biodiversity conservation. Proper mechanization can support scaling up agroforestry practices.

The research outcomes can inform policymakers and government agencies responsible for agricultural development. They can utilize the findings to design supportive policies, incentives, and financial schemes that promote the adoption of appropriate machinery for agroforestry operations. This can contribute to the growth of agroforestry as a sustainable and viable agricultural practice. The provision of support exerts a significant influence among farmers [71]. An agricultural policy that aims to improve access to agricultural machinery must go hand-in-hand with the provision of credit and training, targeting poor and marginalized farmers so that they can sustain agricultural production and ensure food security [72].

Designing effective policies and programs for mechanization in agriculture requires a location-specific analysis [73]. Effective leadership strategies, along with strong, target-

oriented, and pro-farmer policies, are crucial for the successful application of agricultural machinery [69]. Some propositions common in the public debate state that "mechanization leads to unemployment" and "smallholders are unable to obtain benefits from mechanization", but [74] concluded that the mechanization of agricultural operations replaced bullock labor rather than human labor and that the increased utilization of tractors was linked to a significant rise in employment, primarily attributable to their impact on cropping intensity.

It is crucial to have a comprehensive understanding of the economic, agro-climatic, and social factors in order to determine appropriate technological and institutional solutions for each context. By adopting this approach, potential negative effects of mechanization, such as unemployment, can be mitigated. It ensures that mechanization is not artificially imposed in areas where the necessary conditions for its success are lacking [73].

*4.2. Managerial Application*

The research findings provide valuable insights for farmers, extension technicians, and agricultural managers in selecting the most suitable machinery for agroforestry operations based on their specific needs and resources. The analysis assists in making informed decisions regarding machinery investment and optimizing the utilization of available capital. The research outcomes offer a comprehensive understanding of the strengths and weaknesses of different machines for specific agroforestry tasks. This knowledge empowers managers to make informed decisions when selecting machinery that aligns with their production goals, resource availability, and operational requirements. By considering factors such as the size of the farming area, labor availability, and desired outcomes, managers can strategically invest in machinery that maximizes productivity and efficiency. According to [75], project decision making should focus on selecting the optimal investment location in order to maximize economic benefit, utilizing scientific and rational methods.

The analysis of machinery suitability helps managers allocate resources efficiently. By selecting the most appropriate machines, managers can minimize costs associated with labor, fuel consumption, maintenance, and machinery acquisition. This ensures that the available resources are utilized optimally, leading to improved financial performance and profitability [76]. Selecting the right machinery enables managers to streamline agroforestry operations and improve overall efficiency. Machines that are well-suited for specific tasks can reduce the time and effort required to complete them, resulting in increased productivity, timely execution of operations, and the reduction of losses [77]. This not only boosts the yield but also allows for better resource utilization and planning, ultimately enhancing the overall performance of the agroforestry system.

As the field of agroforestry continues to evolve, new techniques, crop varieties, and management approaches may emerge. Managers should consider the versatility and flexibility of machinery options to accommodate future changes and advancements in agroforestry systems. When it comes to policymakers' decisions, the policy can only be realistic and functional if they consider another aspect, according to [78], the voice of the farmers influenced by their decisions. The meticulous selection and investment in machinery based on the findings of this research empowers managers to make informed decisions, administer resources effectively, enhance operational efficiency, promote sustainability, and adapt to changing agroforestry practices.

*4.3. Impact on Sustainable Development*

This study is directly related to the United Nations Sustainable Development Goals (SDGs) in several ways. Specifically, it directly addresses three of the 17 goals, namely, SDG 2 (Zero Hunger), as Agroforestry Systems (AFS) can contribute to sustainable and resilient food production by integrating tree species with crop cultivation [79]. By evaluating the suitability of existing machines for AFS tasks, the study aims to improve the efficiency and productivity of these systems, thereby supporting the goal of achieving food security and promoting sustainable agriculture.

Additionally, it aligns with SDG 8 (Decent Work and Economic Growth) as the current reliance on manual labor in AFS poses physical challenges to small-scale farmers and limits their implementation capacity. By assessing the suitability of different machines, the study seeks to reduce physical effort, increase operational efficiency, and potentially create employment opportunities in the agricultural sector. It also presents new production alternatives to small farmers, resulting in cost savings and increased profitability [67].

Furthermore, the study contributes to SDG 9 (Industry, Innovation, and Infrastructure) as it evaluates the performance of existing machines in agroforestry tasks, aiming to promote the development and adoption of appropriate machinery for AFS. This supports the objective of building resilient infrastructure, promoting inclusive and sustainable industrialization, and fostering innovation for small farmers [80].

*4.4. Future Scope*

In every country's agricultural sector, the process of upgrading agricultural equipment is underway through the development of more sophisticated designs or by leveraging the knowledge and expertise of producers from other nations [81]. There is no proper machine in the market designed specifically for AFSs. This research studied some of the machines available for purchasing, which can be adequate for upgrading the tasks required by AFSs in order to accelerate the working time, reduce the physical effort of farmers, and, mainly, provide greater biomass yield along the harvests.

Future studies should explore developing specific machines for AFSs. In the same direction, studies should explore solar powered brushcutters and harvesters [82–84] as well as automation [85–87]. This research is the beginning of studies regarding agroforestry mechanization. To reach all the benefits that can be provided by AFSs, agriculture in general must be based on it, with larger areas implemented worldwide. This will only be achieved with the development of innovative machines that can efficiently handle tasks unique to agroforestry, such as cutting grass biomass in interrows or managing complex biodiversity in AFSs. According to [88], to achieve higher yields, the most effective approach is to focus on agronomic advancements, with a particular emphasis on optimizing the harvesting process to maximize the actual biomass output. The development of autonomous machines capable of performing precise tasks in AFSs can reduce the physical effort required and increase operational efficiency, paving the way for larger-scale implementation of agroforestry practices.

GRM, as shown in the results chapter, has provided the best cutting systems that resulted in greater amounts of DM, lowest fuel consumption, and minimal raking time. This machine could show even better results if some simple and easy adjustments are made in order to prevent clogging, as suggested by [88], which would reduce HT and, consequently, LC. This would make GRM's results better than RBC's in any other parameter studied here.

There is a growing focus on sustainable agricultural practices. Future developments in mechanization for agroforestry should prioritize sustainability, including reduced environmental impact, energy efficiency, and the use of renewable resources. Integration of eco-friendly technologies and practices into agroforestry machinery can contribute to environmental preservation and resource conservation. Agroforestry represents a progressive approach to sustainable agriculture, aiming to enhance yields, mitigate adverse impacts, and deepen our comprehension of the intricate interplay involved in augmenting food production while reducing harm. By embracing integrated and biodiverse practices, agroforestry offers a promising pathway towards the next stage of sustainable agriculture [89].

The preservation and revival of ecosystems, along with the establishment of sustainable food systems, should be prioritized in the realm of food production. This necessitates the adoption of a proactive and rational management approach as well as fundamental shifts in economic development patterns and production practices. It is imperative to redesign food systems in order to have a neutral or positive environmental footprint, while also guaranteeing healthy nutrition and food safety as well as prioritizing low-impact

environmental strategies [90]. Agricultural machinery is an essential input if sustainable production is to be increased and thereby feed the world population [91].

## 5. Conclusions

The key results of the research are:

- The mini grain reaper machine is optimal for ensuring higher dry matter production at lower costs and providing quick raking time.
- The costal brushcutter is suitable for tasks in small areas.
- The tractor mounted rotary brushcutter is most suitable for large-scale handling, as it is the fastest machine and requires minimal physical effort and labor cost.

It is not feasible to select a machine that is perfectly suited to all of the diverse situations in family farming. The variability observed in the results indicates that each treatment is best suited depending on what is most important for the farmer, taking into account their individual circumstances and their needs when choosing the machine. The decision to choose one of the machines tested in this study should consider an assessment of the farming family's requirements, the size of the working area, the available capital for machinery purchase and maintenance, the available labor force, and the possibility of hiring personnel. Once the factors mentioned above are defined, the suitability observed in this research becomes crucial for decision making.

**Author Contributions:** Conceptualization, G.F.d.M., D.A. and J.d.S.G.S.; methodology, G.F.d.M., L.O.R.F., M.G.B.X., D.A. and J.d.S.G.S.; software, G.F.d.M. and J.d.S.G.S.; validation, D.A., L.O.R.F. and V.F.d.S.-E.; formal analysis, G.F.d.M., D.A., J.d.S.G.S. and D.H.; investigation, G.F.d.M., L.O.R.F., M.G.B.X., D.H., L.S., H.T.d.S. and F.T.d.C.; resources, L.O.R.F.; data curation, G.F.d.M., L.O.R.F., M.G.B.X., D.H., L.S., H.T.d.S. and F.T.d.C.; writing—original draft preparation, G.F.d.M.; writing—review and editing, G.F.d.M. and J.d.S.G.S.; visualization, G.F.d.M.; supervision, L.O.R.F. and D.A.; project administration, G.F.d.M., L.O.R.F. and M.G.B.X.; funding acquisition, L.O.R.F. All authors have read and agreed to the published version of the manuscript.

**Funding:** This research was funded by the Brazilian Agricultural Research Corporation (EMBRAPA), grant number SEG: 20.18.03.031.00.00.

**Institutional Review Board Statement:** Not applicable.

**Data Availability Statement:** For reasons of NDA (non-disclosure agreement) among the institutional partners, confidentiality is required, and, therefore, the data are not made available.

**Acknowledgments:** The authors would like to thank EMBRAPA Meio Ambiente for hosting and for making the experiment possible.

**Conflicts of Interest:** The authors declare no conflict of interest.

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
