# Peer review of "Agricultural Machinery Adequacy for Handling the Mombaça Grass Biomass in Agroforestry Systems"

_agriculture, doi:10.3390/agriculture13071416_

Round 1
Reviewer 1 Report
Introduction must be added recent literatures (2019-2023).research methodology must be presented in an innovative way.sequence is not correct.result part needs more discussion elaboration and comparision.SWOT analysis .future scope ,manegerial and practical application must be added
english must be improved use native english
Reviewer 2 Report
The presented manuscript analyzed the effects of three different cutting machines of Mombaça grass: a mini grain reaper machine (GRM), a coastal brushcutter (CBC) and a tractor-mounted rotary brushcutter (RBC). After three cycles of mowing the grass plots, seven parameters were measured and evaluated in order to describe the observed types of mowing.
In the paper, it is necessary to emphasize the more significant aspects of the scientific and social need for conducting research.
The methodology of the work is described in detail. However, it must be stated which scientific methods were used for the research?
A sentences cannot start with a quote and lowercase letters:
- Lines (51-54). “[15] has noted that agricultural mechanization makes work easier and significantly reduces production costs, while [16] contend that mechanization is likely to have broader impacts on agronomy, the environment, and socioeconomic factors than is commonly recognized.”
- Lines (470-473). [42] also tested GRM in Egypt for bean harvesting. They observed that the machine experienced clogging issues, leading to significant yield losses of over 50%. Higher operating speeds further exacerbated the losses due to the excessive plant load on the cutting bar.
- Lines (532-534). [43] compared the influence of cutting height (CH) on Mombaça grass and observed that the lower the CH, the higher the number of new tillers and the lower the number of regrowths from apical buds emitted during the regrowth process.
Authors are advised not to use the following mathematical symbols in equations, such as: ”* ”; ”^” and ”. ”, etc. Instead of existing symbols, they should use generally accepted mathematical symbols.
In many places in the paper, a period should be used instead of a comma for the numbers listed, see Table 1-4; corrections should also be made in the text itself.
The research results must be compared with the results of other authors.
In the chapter Conclusion before the results, one sentence is missing, e.q. "The key results of the research are:"
References must include multiple sources published in English. The literature presented in this way indicates that the authors did not see the broader context of the research problem.
ISBNs should be deleted from individual references in the literature.
Reviewer 3 Report
Agroforestry Systems are playing an increasingly important role in most countries of the world. The article presented for review refers to the analysis of the work efficiency of three systems of different machines when cutting Mombaça grass. Before possible publication, however, a number of corrections should be made to it:
1. Keywords should not be repeated with the presented title (mombaça grass; agricultural machinery; agroforestry). Make please the appropriate correction.
2. Introduction - topics related to agroforestry systems should be extended with additional literature. Lines 71-74 should be located at the end of the chapter. Lines 75-90 are part of Methodology. Lines 97-100 are part of Conclusions.
3. Subsections 2.1-2.4 should be combined. Lines 123-127 - may be part of the Introduction.
4. Please check the spelling of all units as there are spaces missing in many places, eg line 141 is 3kg when it should be written 3 kg. Please refer to the writting rules of units.
5. Line 243 and others – there is no need to describe equations.
6.2.6.3. Statistical Method - instead of "method" it should be written "analysis".
7. Line 366 - this is not equation 2, but 8.
8. Subchapter 2.6.4. is too short.
9. Results should be separated from Discussion. In fact, discussion was not carried out in this article. It requires consideration at a general level in relation to other ongoing research on similar topics.
10. Please correct row 2 in table 2.
11. Line 532. Please do not start your sentences with a reference number. Put firstly the name of author and then number at the end of the sentence.
12. Decimal digits should be given not after commas, but after dots.
13. Conclusions should contain full sentences referring to the achieved results - no abbreviations should be used.
14. The article should be written in an impersonal form.
Round 2
Reviewer 1 Report
Accepted
Reviewer 2 Report
I agree with all the changes made in the revised version.
Reviewer 3 Report
Please check the writing of "%" because in some cases I see spaces between number and unit. It should be written for example 5%, but not 5 %.